# Bridging the gap between Learning-to-plan, Motion Primitives and Safe Reinforcement Learning

**Piotr Kicki**[1,2]**, Davide Tateo**[3]**, Puze Liu**[3]**, Jonas Guenster**[3]**, Jan Peters**[3]**, Krzysztof Walas**[1,2]

[1]IDEAS NCBR, Warsaw, Poland

[2]Institute of Robotics and Machine Intelligence, Poznan University of Technology, Poland

[3]Department of Computer Science, Technische Universitat Darmstadt, Germany

piotr.kicki@ideas-ncbr.pl

https://pkicki.github.io/CNP3O/

**Abstract:** Trajectory planning under kinodynamic constraints is fundamental for advanced robotics applications that require dexterous, reactive, and rapid skills in complex environments. These constraints, which may represent task, safety, or actuator limitations, are essential for ensuring the proper functioning of robotic platforms and preventing unexpected behaviors. Recent advances in kinodynamic planning demonstrate that learning-to-plan techniques can generate complex and reactive motions under intricate constraints. However, these techniques necessitate the analytical modeling of both the robot and the entire task, a limiting assumption when systems are extremely complex or when constructing accurate task models is prohibitive. This paper addresses this limitation by combining learning-to-plan methods with reinforcement learning, resulting in a novel integration of black-box learning of motion primitives and optimization. We evaluate our approach against state-of-the-art safe reinforcement learning methods, showing that our technique, particularly when exploiting task structure, outperforms baseline methods in challenging scenarios such as planning to hit in robot air hockey. This work demonstrates the potential of our integrated approach to enhance the performance and safety of robots operating under complex kinodynamic constraints.

**Keywords:** safe reinforcement learning, motion planning, motion primitives

## 1 Introduction

Nowadays, robots are capable of complex dynamic tasks such as table tennis [1, 2], juggling [3, 4] or diabolo [5], and play sports such as tennis [6] or soccer [7]. Current planning and learning methods are sufficient for most of these tasks, as the robot's movement is relatively free in the workspace, and they are not required to comply with stringent tasks, hardware, and safety constraints. However, these requirements become fundamental if we want to deal with real robotics tasks in unstructured environments in the real world. Therefore, the lack of competitive techniques to efficiently plan trajectories in unknown and unstructured environments under constraints strongly limits the applicability of modern robotics and learning frameworks to tasks beyond the lab setting.

To fix this gap, Altman [8] introduced the Constrained Markov Decision Processes (CMDP) framework, and, based on this setting, researchers in machine learning developed the Safe Reinforcement Learning (SafeRL) techniques to efficiently solve the CMDP problem without full knowledge of the environment. However, these methods are not able to scale effectively to complex tasks. Furthermore, since most of these approaches learn black-box approximations, they do not allow effective exploitation of domain knowledge.

Motion Primitives (MPs) are a technique that enables efficient encoding of domain knowledge in the policy not fully exploited in the SafeRL context. Interestingly, these approaches plan full trajectories, allowing the agent to check for the safety of the whole trajectory before executing it, preventing the agent from reaching states that may heavily violate the constraints. However, given that MPs

8th Conference on Robot Learning (CoRL 2024), Munich, Germany.

allows encoding the safety features in the trajectory, the literature lacks general approaches to deal with a general set of constraints. Unfortunately, this forces the user to rely heavily on hand-crafting MPs, possibly resulting in suboptimal solutions.

A valid alternative to the methods above is to exploit learning-to-plan methods that can generate full trajectories, as in the MP setting, while imposing constraints during the planning time or even in the learning process. However, learning-to-plan approaches require the full knowledge of the task being optimized. This assumption is very different from the SafeRL setting, where we assume to have access only to environment rollouts.

In this paper, we draw connections between these aforementioned fields and we show how to extend the learning-to-plan methods to exploit the knowledge of the constraints without requiring full knowledge of the environment. We extend the framework presented in [9] to the Reinforcement Learning (RL) setting, resulting in a hybrid method that shares many common ideas with the SafeRL, MP, and learning-to-plan approaches. In particular, the contributions of this paper are i. a novel algorithm to learn how to generate MP-based trajectories under known constraints; ii. an analysis of different MPs, where we show that the B-splines are particularly useful for learning under constraints; iii. practical guidelines on how to properly impose domain knowledge on MPs in the learning-to-plan setting.

We evaluate our approach in two challenging tasks, i.e., moving a heavy vertically-oriented object with a manipulator and a robotic air hockey hitting task. The first task is challenging as it pushes the limits of robot actuators, while the second task requires learning a highly dynamic motion under complicated constraints and sensitive objective functions. Finally, we show that we could deploy the proposed method in a real robot air hockey setup.

**Related work**

The literature on safety is quite broad, and there are many different SafeRL approaches that tackle the CMDP problem [8, 10]. The first and most popular technique to solve this problem is the lagrangian relaxation approach [8, 11, 12, 13, 14, 15, 16, 17], where the original task objective is mixed trough a lagrangian multiplier with the constraint cost. Other alternative solutions instead rely on different ideas from the optimization literature, such as state augmentation [18], the trust region methods [19, 20] and the interior point approach [21]. An alternative formulation of the problem is based on more control-theoretic insights. These approaches are based either on Lyapunov functions [22, 23, 24], Control barrier functions [25, 26, 27, 28] or reachability analysis [29, 30, 31, 32]. The last category of solutions to safety problems is shielding techniques that correct potentially dangerous actions to be applied in the system [33, 34, 35, 36, 37, 38, 25, 39, 40, 41]. Some approaches in these two last categories can guarantee safety at every step of the learning process but require prior knowledge of system dynamics, backup policies, or unsafe interaction datasets.

A very classical approach to introduce safety into robot actions is motion planning [42]. However, typically classical planning methods struggle to meet the real-time requirements for complex tasks. Thus, we observe the growing interest in the motion planning community for exploiting learning to improve and speed-up planning [9, 43, 44, 45, 46]. Many approaches that apply learning to motion planning utilize learning to bias the prediction of the next segment of the solution [43, 47, 45]. However, methods of this type struggle to generate feasible dynamic trajectories for complex problems with challenging constraints [9]. An interesting alternative to combine planning and learning is so-called *planning-as-inference*, an example of which may be the use of diffusion models to optimize whole trajectories [46] or inferring the whole motion planning problem solutions with a neural network [44, 48]. In this spirit, authors of [9] introduced a machine learning-based method for planning trajectories that satisfy a variety of constraints to ensure the system safety. However, the approach introduced in [9] requires the differentiability of the task loss function, which is not the case for many interesting real-world problems.

A very important aspect of learning how to generate trajectories is their representation. While it is possible to follow the auto-regressive approach [43, 47], methods that utilize more structured

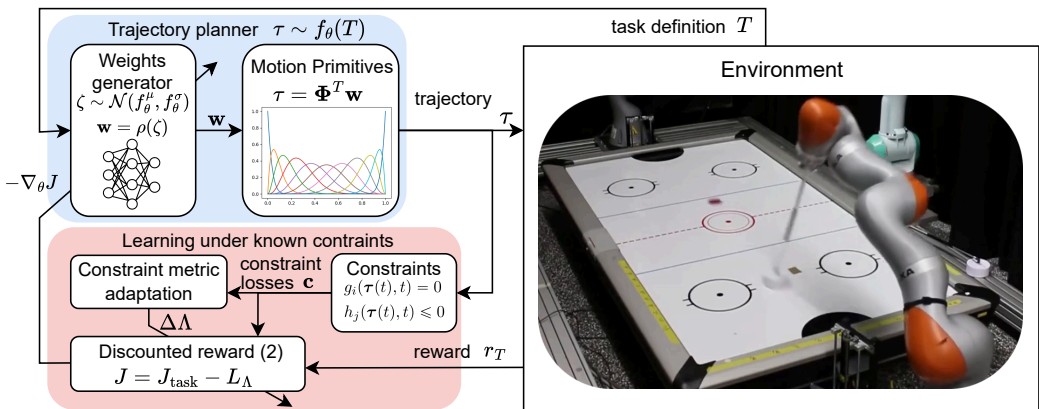

Figure 1: Overview of the proposed constrained trajectory generation method.

trajectory representations seem to offer more benefits, like compactness, reduced planning time and boundary conditions satisfaction [9, 48, 44, 49, 50]. One of the most common approaches for trajectory representation in learning-based solutions are MPs [49, 51, 50, 48, 52], as they offer a very flexible and compact representation. To the most widespread MPs belongs Probabilistic Movement Primitives (ProMP) [49] and Dynamic Movement Primitives (DMP) [51], which benefits were recently combined in Probabilistic Dynamic Movement Primitives (ProDMP) [50], allowing for some simple boundary constraints satisfaction and efficient computation, which is also possible with the use of MPs presented in [48]. However, by far the most composite and flexible solution was introduced in [9], where B-splines were used to construct a trajectory. This representation allows for imposing boundary conditions on the trajectory and its derivatives and introduces great flexibility in terms of the trajectory timing.

This paper, builds upon the approach introduced in [9]. We show that trajectory representation from [9] is in fact a MP, and we extend the learning-to-plan method that uses them to RL setting via a hybrid approach that bridges the gap between learning-to-plan, MPs and SafeRL.

## 2  Constrained Reinforcement Learning with Motion Primitives

Our problem is to find a sampling distribution $\pi$ over trajectories $\boldsymbol{\tau}$ maximizing the expected cumulative reward while satisfying the safety constraints $g, h$ over the whole trajectory. The performance objective is defined on a distribution of tasks $\mathcal{T}$ parameterized with a task definition $T$. The resulting optimization problem is

$$\underset{\pi}{\arg\max} \quad \underset{T \sim \mathcal{T}}{\mathbb{E}}\left[\underset{\boldsymbol{\tau} \sim \pi}{\mathbb{E}}\left[\sum_{t=0}^{t_{\boldsymbol{\tau}}} \gamma^t r_T(s_t, a(\boldsymbol{\tau}(t)))\right]\right] \quad \text{s.t.} \quad \begin{aligned} &g_i(\boldsymbol{\tau}(t), t) = 0 \; \forall t, \forall i \in \{1, \ldots, N\}, \\ &h_j(\boldsymbol{\tau}(t), t) \leqslant 0 \; \forall t, \forall j \in \{1, \ldots, M\}, \end{aligned} \quad (1)$$

where $\sum_{t=0}^{t_{\boldsymbol{\tau}}} \gamma^t r_T(s_t, a(\boldsymbol{\tau}(t)))$ is the discounted task reward $J_{\text{task}}$. We assume that the action $a$ is computed by a controller based on the trajectory representation $\boldsymbol{\tau}(t)$ and the current time $t$.

To solve this problem, we propose to generate search distribution $\pi = f_\theta(T)$ using a function $f_\theta$, parametrized with $\theta$, based on the given task definition $T$. We formalize this function as a linear combination of MPs with weights computed with a nonlinear transformation $\rho$ of the samples drawn from the normal distribution, in which mean and standard deviation are determined by a neural network. Generated trajectories are evaluated in the environment, and based on the trajectory and the reward from the environment, we optimize the neural network weights $\theta$ to maximize the task reward and minimize the constraints violations. The overview of the proposed solution is presented in Figure 1, while we discuss its core components in the next sections.

### 2.1  Learning under known constraints

This section, describes the proposed solution for solving the constrained optimization problem defined in (1). One of the most common approaches to address this challenge is the method of Lagrange

multipliers [8, 16, 17]. Inspired by this approach, we propose to relax the constraints, assuming some acceptable violation budget, and include them in the objective with learnable scaling factors. In this paper, the optimization with the extended objective is interleaved with the adaptation of the Lagrange multipliers associated with specific constraints, which can be interpreted as learning the metric of the constraint manifold. We exploit the knowledge about the considered system constraints and the fact that typical robot constraints are differentiable w.r.t. trajectory $\boldsymbol{\tau}$ to optimize the neural network to satisfy them using their analytical gradient. Moreover, we show that it is possible to include a range of different constraints without aggregating them into a single constraint cost function, which enables better handling of multiple constraints. In the following, we present the Constrained Neural motion Planning with PPO (CNP3O) algorithm (see Algorithm 1), and we show how it can be derived by extending the Constrained Neural motion Planning with B-splines (CNP-B) approach [9] to the RL setting.

First, following [9], we transform both equality $g_i(\boldsymbol{\tau}(t), t) = 0$ and inequality constraints $h_j(\boldsymbol{\tau}(t), t) \leqslant 0$ into inequality constraints of the form $c_i(\boldsymbol{\tau}(t), t) \leqslant \bar{C}_i$, where $c_i(\boldsymbol{\tau}(t), t)$ represents the $i$-th constraint violation of the trajectory $\boldsymbol{\tau}(t)$ and $\bar{C}_i$ is the assumed $i$-th constraint violation budget. Then, we need to incorporate these constraints into the task objective. To do so, we rewrite the objective function from (1) into

$$J = \mathop{\mathbb{E}}_{T \sim \mathcal{T}} \left[ \mathop{\mathbb{E}}_{\zeta \sim f_\theta(T)} \left[ \sum_{t=0}^{t_\tau} \gamma^t r_T(s_t, a(\rho(\zeta(t)))) \right] - \boldsymbol{c}^T(\rho(f_\theta^\mu(T)), t) \Lambda \boldsymbol{c}(\rho(f_\theta^\mu(T)), t) \right], \qquad (2)$$

where $f_\theta^\mu(T)$ is the true mean of the distribution induced by the neural network $f$ for a given task $T$, $\rho$ is a nonlinear transformation of the samples $\zeta$ (for more details see Appendix C), $\Lambda = \mathrm{diag}(\lambda_1, \lambda_2, \ldots, \lambda_{N+M})$ is the diagonal matrix of the Lagrange multipliers and $\boldsymbol{c}^T(\rho(f_\theta^\mu(T)), t) \Lambda \boldsymbol{c}(\rho(f_\theta^\mu(T)), t)$ is the manifold loss $L_\Lambda$. In this way, we decouple learning how to solve a task from learning how to satisfy the constraints and allow one to optimize the constraint satisfaction against the neural network weights $\theta$ directly, without the need for sampling. By doing so, we are not penalizing the constraints violation done by the samples of the trajectory distribution but only its mean. Therefore, to ensure safety in the real robot deployment, we assume that the mean trajectory will be used, while for learning on the real robot, one may not execute trajectories that violate the constraints too much. We explicitly allow for imposing multiple constraints in a decoupled way, such that each of them has its own scaling factor $\lambda_i$. To ensure the positiveness of these scaling factors, we parameterize each of them with $\eta_i$ using $\lambda_i = \exp(\eta_i)$. These scaling parameters $\eta_i$ are updated based on the mean constraints violations observed in the previous set of episodes, which can be defined by $\Delta\eta_i = \alpha \log \left( \frac{c_i + \beta \bar{C}_i}{\bar{C}_i} \right)$, where $\alpha > 0$ is the constraint learning rate and $0 < \beta < 1$ bounds the rate of the $\eta_i$ decline.

The remaining part of the discounted reward $J$ is related to solving the task. To evaluate the sample task reward, we first generate the normal distribution $\mathcal{N}(f_\theta^\mu(T), f_\sigma^\mu(T))$ using neural network $f_\theta$, which predicts its mean $\mu$ and standard deviation $\sigma$ for a given task $T$. Then, we sample from $\zeta \sim \mathcal{N}(f_\theta^\mu(T), f_\sigma^\mu(T))$, process these samples with the samples transformation function $\rho(\zeta)$, and evaluate generated trajectories $\boldsymbol{\tau}$ in the considered environment. Finally, we use the obtained accumulated rewards from the simulated episodes to optimize the neural network weights $\theta$, using an episodic version of the Proximal Policy Optimization (PPO) algorithm [53] due to its simplicity (details of the episodic PPO are provided in Appendix F).

## 2.2  Motion Primitives for Safe Reinforcement Learning

One of the most important design decisions in the case of learning how to generate plans is the representation of the trajectories that the planner will generate. MPs [48, 50] are very popular, general, and flexible trajectory representations that can be used for planning. They can be, in general, defined by $\boldsymbol{q}(s) = \boldsymbol{\Phi}(s)\boldsymbol{w}$, where the robot configuration $\boldsymbol{q}(s) \in \mathbf{R}^{n_q}$, for given value of the phase variable $s$, is computed as a product of basis functions $\boldsymbol{\Phi}(s) \in \mathbf{R}^{1 \times n_b}$, evaluated at $s$, and the weights vector $\boldsymbol{w} \in \mathbf{R}^{n_b \times n_q}$. This formulation may describe diverse MPs, such as ProMP [49], ProDMP [50], Residual Trajectory Primitives (RTP) [48], just by defining basis functions $\boldsymbol{\Phi}$ differently or adding

some biases to the weights $\boldsymbol{w}$. To interpret these MPs as trajectories, we need to transform the dependency on the phase variable $s$ into dependency on time $t$. In the literature, it is typical to do it directly by assuming that $s = t$, or by a linear scaling, i.e., $s = \frac{t}{T_s}$, where $T_s > 0$ is the time scaling factor. Each of the aforementioned MPs offers different useful properties from the motion planning point of view, such as the guarantee of connecting the initial $\boldsymbol{q}_0$ and target $\boldsymbol{q}_d$ configurations. However, we argue that much more may be offered by using the B-spline-based MPs. In this work, we show that the trajectory representation proposed in [9] can be viewed as MP and highlight its benefits over the existing MPs-based approaches.

First, let us note that any spline function of order $n$ can be represented as a linear combination of $n$-th order B-splines, i.e. $\boldsymbol{q}(s) = \sum_i \boldsymbol{w}_i B_{i,n}^{\boldsymbol{k}}(s)$, where $B_{i,n}^{\boldsymbol{k}}$ is the $n$-th order B-spline basis function defined between $k_i$ and $k_{i+n+1}$, where $\boldsymbol{k}$ is a vector of knots that define domains of the B-spline basis functions. In general, the vector of knots $\boldsymbol{k}$ may be an arbitrary non-decreasing sequence that partitions the domain of the represented function, such that its changes affect the shape of the MPs, which would have to be then recomputed every time. To avoid this, we propose fixing the knots vector and limiting it to the range of $s \in [0; 1]$. Thanks to this, we can drop the dependency on the knot vector $\boldsymbol{k}$ and describe B-splines as MPs $q(s) = \boldsymbol{\Phi}(s)\boldsymbol{w}$, where $\boldsymbol{\Phi}(s) = [B_1(s) \quad B_2(s) \quad \cdots \quad B_n(s)]$. Moreover, it is possible to precompute all of the basis functions in advance for a range of the phase variable values $\{0, \frac{1}{n_s-1}, \frac{2}{n_s-1}, \dots, 1\} \in \mathbf{R}^{n_s}$ and create a tensor $\boldsymbol{\Phi} \in \mathbf{R}^{n_s \times n_b}$, which allows for computing the movement as a matrix-vector product, similarly like it can be done for ProMP [49] and ProDMP [50]. Although the choice of knot vector $\boldsymbol{k}$ may be arbitrary, we propose making subsequent knots equidistant to distribute the basis functions equally across the domain. Moreover, by fixing the first and last $n + 1$ knots to 0 and 1, respectively, we can guarantee that the generated path starts at the point defined by $\boldsymbol{w}_1$ and ends in $\boldsymbol{w}_{n_b}$.

Finally, we must transform the phase variable $s$ into the time $t$ to generate a trajectory. Therefore, we introduce another B-spline function $r(s) = \left(\frac{dt}{ds}\right)^{-1}(s)$, as done in [9]. This flexible time scaling allows us to flexibly control the derivatives of the resultant trajectory w.r.t. time and the overall trajectory duration, which is not so straightforward for the methods proposed in [49, 50, 48]. Moreover, this time parameterization allows one to directly impose the boundary constraints on the initial and target velocities, accelerations, and higher order derivatives up to the $d$-th derivative by analytically setting the $d + 1$ boundary weights of the configuration. To the best of our knowledge, this flexibility in imposing boundary conditions is not present in any existing MP framework.

Imposing boundary conditions is an important feature of the trajectory representations as it allows for connecting the subsequent trajectories, which is particularly important to concatenate trajectory segments, enable online replanning, and may be useful in incorporating some prior knowledge into the designed solution. An example of this feature is a task that features reaching a known goal. Then, we can easily impose this goal configuration as a prior for the solutions returned by the planner.

---

**Algorithm 1** CNP3O

1: **for** $k \leftarrow 1$ to $N_{\text{epochs}}$ **do**
2:     **for** $i \leftarrow 1$ to $N_{\text{episodes}}$ **do**
3:         Sample task $T$ from distribution $\mathcal{T}$
4:         Sample $\zeta$ from the distribution $\mathcal{N}(f_\theta^\mu(T), f_\theta^\sigma(T))$ and compute the sample trajectory $\boldsymbol{\tau} = \boldsymbol{\Phi}\rho(\zeta)$
5:         Evaluate $\boldsymbol{\tau}$ in the environment to get $J_{\text{task}}$.
6:         Compute the value function $V_\psi(T)$ of task $T$
7:         Store $(\zeta, J_{\text{task}}, f_\theta^\mu(T), V_\psi(T))$
8:     **for** $i \leftarrow 1$ to $N_{\text{fits}}$ **do**
9:         **for** $j \leftarrow 1$ to $N_{\text{batches}}$ **do**
10:             Sample a batch of $N_{\text{episodes}}/N_{\text{batches}}$ elements from the stored data
11:             Compute task loss $L_{\text{task}}$ based on the $\zeta$, $J_{\text{task}}$ and $V(T)$ using (8)
12:             Compute manifold loss $L_\Lambda$ based on the $\rho(f_\theta^\mu(T))$
13:             Update the policy network weights $\theta$ based on the gradient of $L_{\text{task}} + L_\Lambda$
14:             Update the value network weights $\psi$ based on the gradient of $(J_{\text{task}} - V_\psi(T))^2$
15:         Update manifold metric $\Lambda$

---

Thanks to the ability to impose boundary conditions also on the derivatives, we can incorporate strong priors that enable ending the motion with some predefined velocity, useful in tasks that require hitting, moving, or tossing objects. This is also important for connecting trajectories smoothly, e.g., making $\dddot{q}(t)$ continuous we can ensure the smoothness of controls, despite the change of the trajectory segment. Thanks to this capability, we can generate both single trajectories and smoothly connected sequences of trajectories passing through the via-points. Further discussion on imposing prior knowledge on MPs-based trajectories is provided in Appendix D.

## 3   Experimental Evaluation

In this section, we evaluate the performance of our proposed method in two tasks defined in [9], using 10 seeds. In both environments, the objective is to control a Kuka Iiwa 14 manipulator. Our experimental evaluation has two objectives. First, we want to evaluate the benefit of our framework against classical SafeRL approaches. For this reason, we consider following important baselines in the area, Acting on the TAngent Space of the COnstraint Manifold (ATACOM) [39, 41, 54] – a model-based approach, model-free methods PPO-Lagrangian (PPO-Lag) and TRPO-Lagrangian (TRPO-Lag) [55], and a projection based extension of TRPO-Lag – Projection-Based Constrained Policy Optimization (PCPO) [56]. All of these approaches learn a classical neural policy to control the robot. However, ATACOM ensures safety by exploiting the model of the robot and the constraints, PPO-Lag and TRPO-Lag implement a model-free lagrangian optimization technique, while PCPO additionally projects the policy on the constraint set. The second objective of the conducted experiments is to evaluate which MP is more adequate for the learning-to-plan setting. Therefore, we compare the B-splines [9], ProMP [49] and ProDMP [50].

**Heavy object task**   The first task consists of moving a heavy object with a manipulator from one block to another while avoiding collision, fulfilling torque limit requirements, and maintaining vertical orientation. In this task, kinodynamic constraints are particularly important due to the heavy mass mounted on the robot's end-effector, pushing the required torque commands to the limit.

The results of this experiment are reported in Figure 2ab. The results show that our method outperforms all other approaches in both settings, independently of the use of prior knowledge. We could not obtain good learning performance with PPO-Lag, TRPO-Lag and PCPO. We believe that the combination of kinodynamic constraints and orientation constraints is too strict for both Lagrangian-style approach and projection-based one. Instead, the approaches using MPs obtain satisfactory learning results. In particular, the B-spline representation (CNP3O) can obtain faster and more stable learning results than the alternative MPs. We argue that the B-spline representation allows us to decouple the learning of the geometric path from the controlled execution's speed and, consequently, to deal more easily with different constraints with conflicting requirements. Using MPs is straightforward to introduce prior information into the task, e.g., force the target position for each planning setting. Exploiting prior information allows us to learn the task faster than relying on end-to-end methods. The prior information setting widens, even more, the performance gap between the B-Spline representation and other competing MP parameterizations, presumably due to the ability to enforce zero joint velocity and acceleration at the end of the motion.

**Robot Air Hockey Hitting**   The second task is to teach a robot manipulator to score goals in the robot air hockey setting. Here, the constraints are mainly collision constraints preventing the robot

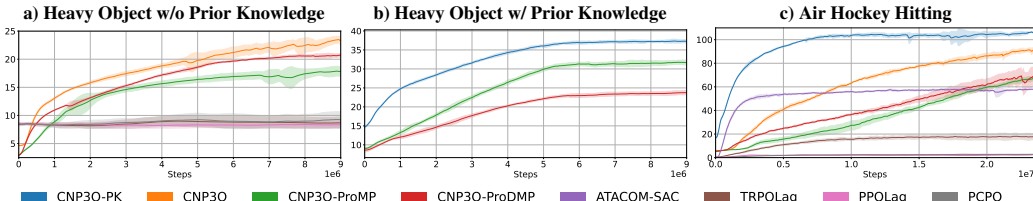

Figure 2: Learning curves (reward w.r.t. number of simulation steps) for the: (a) heavy object task without prior knowledge, (b) with prior knowledge, and (c) air hockey hitting task.

from colliding with the table and the constraint of maintaining the end effector on the table surface. Notice that the policy must control all the joints of the robot, therefore maintaining the surface is extremely challenging without prior knowledge.

We can see the performance of the algorithms in Figure 2c. Here, it is clear that our methodology outperforms all the baselines. In particular, unsurprisingly, the PPO-Lag, TRPO-Lag and PCPO algorithms cannot achieve good performance. Also, in this case, we argue that this poor performance is mainly due to complex constraints that one side cannot be handled effectively without prior knowledge, and on the other side limit the exploration of the projection-based method. In turn, ATACOM quickly converges to a suboptimal solution. This approach is much more suited for learning this task as it exploits prior knowledge. However, ATACOM uses Soft Actor Critic (SAC) as the underlying learning algorithm: stepwise exploration and automatic entropy tuning allow for very fast and effective learning; however, it is often prone to premature convergence. In comparison, our learning method combined with MPs converges to much more performant solutions. We argue this is due to the trajectory-based exploration in a lower-dimensional representation space, which allows more meaningful exploration and easy correction of the behaviors. Regarding the trajectory representation, we also observe in this task that the more flexible B-splines approach allows our method to reach optimal results faster. Furthermore, by adding prior knowledge, our learning speed is comparable with ATACOM, allowing us to reach higher values of the task objective, primarily due to faster-hitting behaviors. We argue that if prior knowledge is fundamental for task safety, exploiting it to make learning more effective is straightforward, particularly if the framework allows easy encoding of the task information.

In Figure 3, we present the detailed metrics of the simulated hitting behaviors at the end of the training. We evaluate the trajectories in terms of success rate, expected discounted return of the mean of the search distribution, maximum puck velocity, and constraint violations in terms of joint and end-effector violations. We observe that CNP3O achieves higher performance and less variant results in all the metrics. In particular, CNP3O with Prior information achieves a scoring ratio close to 100%, and without prior information, only ATACOM outperforms CNP3O. However, it is worth noting that the ATACOM hitting is significantly slower. Moreover, none of the remaining SafeRL baselines can get close to the CNP3O and ATACOM, even they are allowed to make bigger constraints violations (see table constraint violations for PPO-Lag, TRPO-Lag and PCPO and Appendix A.2 for more detailed explanation) Regarding constraints violations, all the MP approaches and ATACOM can maintain the end effector on the table surface quite accurately. The highest joint velocity violations are achieved by CNP3O-based methods, however, the scale of the violations is relatively small and it is primarily caused by the very fast motions at the edge of the robot capabilities.

Finally, we test the zero-shot capabilities of our approach in the real-world robot air hockey setting. We evaluate both the CNP3O and the ATACOM policies for 100 hitting attempts. Looking at the metrics presented in Figure 4, we observe significant performance loss compared with the simulated task, particularly regarding success rate. This drop happens since our learning in the simulated environment does not employ any domain randomization technique. Thus, the big sim-to-real gap is due to the unmodelled disturbances of the air hockey table, imperfect controllers, and delays in both perception and command. Nevertheless, the deployed policy can still hit strongly the puck and score some goals. Compared with ATACOM zero shot deployment, we score fewer goals, however

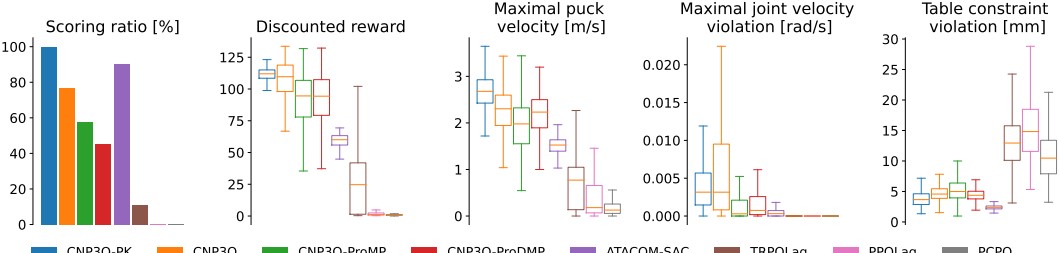

Figure 3: Statistical analysis of the considered approaches on the simulated Air Hockey hitting task.

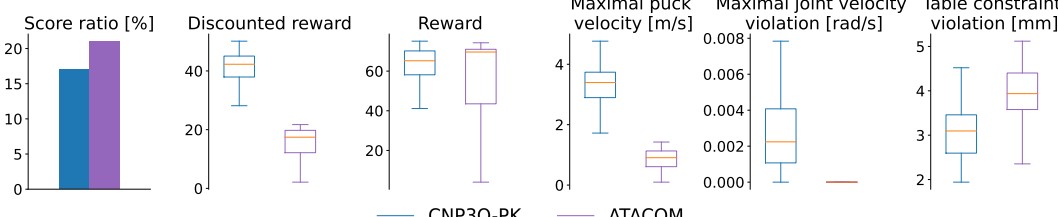

Figure 4: Statistical comparison of CNP3O and ATACOM on the Air Hockey hitting with real robot.

CNP3O achieves much higher total reward, mainly due to much faster hitting behaviors, as clearly shown in the boxplots. It is important to notice that our approach in the real world can obtain lower z-axis violations. We argue that this unexpected result can be attributed to the sampling of entire trajectories, instead of single actions at each timestep. Indeed, this makes our approach less sensitive to the observation noise, which is effectively filtered by the low-level control loop. Another issue for our method not so present in the behaviors learned by ATACOM, is that the mallet occasionally flips during trajectory execution. The metrics suggest that fast movements in combination with small z-axis violations cause this problem.

**Limitations** Our methodology has some limitations compared to standard SafeRL algorithms. First of all, we require the knowledge of the constraints. However, the algorithm can be easily adapted to work with samples from the environment. The second limitation is that our approach is relatively more data-hungry than classical RL approaches as it learns from the entire episodes instead of individual steps. This can be limited by exploiting recent advances in the episodic RL [57] and initializing the solution with imitation learning, speeding up the initial learning phases. Furthermore, our algorithm for MP learning is quite simple and may benefit from modern learning techniques [57] for MP or better trust-region approaches [58]. Finally, while it is possible to adapt the method to learn directly in the real world, this comes at the cost of being able to evaluate the full trajectories beforehand and preventing the execution of trajectories that violate safety constraints too much. Another alternative is to improve the zero-shot transfer of the policy on the robot. This could be done by including domain randomization in the training process or by adding replanning as done in [9]. However, both these approaches require even more samples for the training.

## 4  Conclusion

In this work, we bridge the gap between the SafeRL, MP, and learning-to-plan. The proposed solution combines the strengths of these approaches and outperforms state-of-the-art SafeRL algorithms on two challenging robotic manipulation tasks with a variety of constraints. Our method utilizes the knowledge of the constraints and robot dynamics to improve learning efficiency but, unlike learning-to-plan methods, does not require full knowledge of the environment. Moreover, it leverages MPs to reduce the dimensionality of the action space, facilitate reasonable exploration, and incorporate prior knowledge in the form of boundary conditions. Finally, the proposed solution learns how to generate trajectories satisfying a rich collection of constraints thanks to the introduced learning algorithm inspired by the techniques exploited in SafeRL.

In our experiments, we show that CNP3O allows the robot to perform significantly more dynamic motions while ensuring only minimal violations of the constraints, comparable to much more conservative state-of-the-art approaches. Our real robot evaluation in robot air hockey shows the ability of the introduced method to maintain a safety level comparable to the one observed in simulation and transfer to complex real-world robotics tasks in a zero-shot manner. Last but not least, we show that our proposed MP parametrization is more suitable than existing MPs in case of constrained optimization and also allows the designer to encode more prior knowledge in the learning process, such as desired configuration, velocity, acceleration, and higher order derivatives. This is particularly important for smoothly composing sequences of trajectories together to solve even more complex tasks, which we see as a potential direction for future research.

**Acknowledgments**

This paper is partially supported by Foundation for Polish Science (FNP) and by the German Federal Ministry of Education and Research (BMBF) within the collaborative KIARA project (grant no. 13N16274). This paper was created with the use of the infrastructure of the Poznan Supercomputing and Networking Center.

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

# Appendix

## A  Environments

In this section, we would like to introduce the details of the environments used to evaluate the method introduced in the paper and the baselines. Both of the considered simulation environments were implemented using the MuJoCo physics simulator [59].

### A.1  Heavy Object

The Heavy Object environment is heavily inspired by the motion planning task of moving a heavy vertically oriented object introduced in [9]. In this task, the objective is to control a Kuka Iiwa 14 manipulator holding a heavy box weighing 12kg, such that the box, which is initially placed on one pedestal, is moved to some position on the second pedestal. The main difficulty of this task stems from the fact that the box is pushing the manipulator to its payload limit, which may result in exceeding the maximal torque that can be applied to the robot's joints. We make this task even more challenging by (i) adding constraints on the vertical orientation of the handled object, (ii) preferring faster motions due to the discount factor $\gamma = 0.99$, and (iii) minimizing the sum of the torques applied. Finally, we require both the robot and the object to not collide with the pedestals. A visualization of this task is presented in Figure 5.

A single episode is defined by the initial joint configuration of the robot, which is computed by the inverse kinematics based on the randomly drawn initial position of the handled object placed on top of the first pedestal, and the desired end position of the object placed on top of the second pedestal. The episode horizon is set to 100 steps, each consisting of 20 intermediate 1ms-long steps, which gives 2 s for the entire episode.

In this task, the reward function is meant to encourage minimizing the distance to the goal pose, stopping the robot after reaching the goal, and not using too much energy. To achieve this we proposed the following reward function:

$$R = \frac{1}{10d + 1} + \mathrm{I}(d < 0.01)\frac{0.01}{\|\dot{q}\| + 0.01} - 10^{-6}\|\boldsymbol{m}\|^2, \tag{3}$$

where $d$ is the Euclidean distance between the current and desired position of the heavy object, I is an indicator function that is 1 if the argument is true and 0 otherwise, while $\boldsymbol{m}$ is the vector of joint torques.

Besides the reward function, we also defined several constraints, which are presented in Table 1. The vertical orientation constraint is defined by the element $(2, 2)$ of the handled object rotation matrix $R^o$. In turn, the collision loss is the sum of the collision between all considered collision bodies. We consider collisions between the robot body and pedestals, as well as, the handled heavy object and pedestals. For simplicity, we approximate the robot body by a sequence of 14cm radius balls

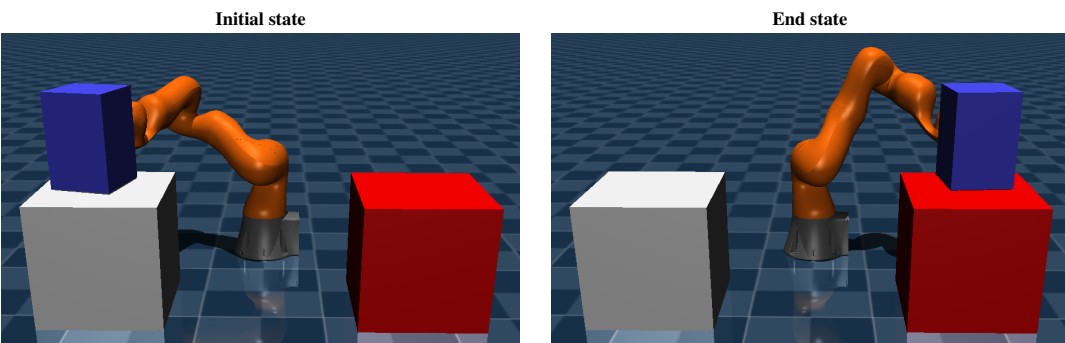

Figure 5: Visualization of the task of moving a heavy object.

located along the kinematic chain, no further than 10cm apart from each other. The value of the collision$(i, j)$ is the analytically computed depth of the penetration. Moreover, there is an implicit constraint imposed on the maximal torques implemented by saturation of the control signals that are possible to be applied to the environment.

**Task definition.** As explained in the main paper, in our setting we assume to have access to a task definition vector $T$. To fully describe this manipulation task, the task description vector contains the initial position and velocity of the robot and the desired pose of the handled object. Notice that this information is the minimal required set to perform the desired motion in a multitask setting.

Table 1: Definition of the constraints in the Heavy Object task.

| No. | Name | Definition |
|---|---|---|
| 1-7 | Joint positions | $|q| \leqslant [2.97, 2.09, 2.97, 2.09, 2.97, 2.09, 3.05]$ |
| 8-14 | Joint velocities | $|\dot{q}| \leqslant [1.48, 1.48, 1.75, 1.31, 2.27, 2.36, 2.36]$ |
| 15 | Vertical orientation | $1 - R_{2,2}^{ho}$ |
| 16 | Collisions | $\sum_{i,j} \text{collision}(i, j)$ |

## A.2   Air Hockey Hitting

In this environment, the main objective is to hit the air hockey puck located on the air hockey table in such a way that it reaches the opponent's goal. The main difficulty of this task comes from the use of a general-purpose 7DoF manipulator (Kuka Iiwa 14) with a long end-effector ending with a mallet. This is particularly challenging due to the constraints that are put on the end-effector, i.e. remaining on the table plane throughout the whole movement, and the requirement to hit as fast as possible and, at the same time, very accurate. The considered setup is presented in Figure 6.

The considered Air Hockey Hitting environment is a slightly adjusted version of the environment *AirHockeyHit* introduced in the Air Hockey Challenge [60]. We kept the original environment in an unchanged form, except for the slightly bigger set of puck initial positions ($[-0.65, -0.25] \times [-0.4, 0.4] \rightarrow [-0.7, -0.2] \times [-0.35, 0.35]$), smaller initial puck velocities range ($[0, 0.5] \rightarrow [0, 0.3]$), initial robot configuration fixed to a single one ($q_0 = [0, 0.697, 0, -0.505, 0, 1.929, 0]$) and shorter episode horizon ($500 \rightarrow 150$ steps). Also, the reward function is the same as in the *AirHockeyHit* environment. For non-absorbing states it gives a reward of $1.5 \, \text{clip}(\dot{x}_p, 0, 3)$, where $\dot{x}_p$ is the puck velocity in $x$ axis, if the puck is in the opponents half of the table. It also encourages the robot's end-effector to get closer to the puck, by rewarding decrease of the distance between them multiplied by a factor of 10, if they were never that close before. Moreover, depending on the type of the absorbing state a different reward, scaled by the $\frac{1-\gamma^h}{1-\gamma}$, where $h$ is the environment horizon, is given. In case of scoring the goal a reward $r = 1.5 - 5 \, \text{clip}(|y_p|, 0, 0.1)$ is awarded, for reaching the opponent band $r = 2(1 - 2 \, \text{clip}(|y_p| - 1, 0, 0.35))$, and in case of hitting the left or right band (from the player perspective) $r = 0.3 - 0.3 \, \text{clip}(l - x_p, 0, 1)$, where $(x_p, y_p)$ is the puck position and $l$ is the length of the table.

Also in the case of this task, we require the satisfaction of both joint position and joint velocities limits and impose the torque constraints by restricting the actuation range. Besides them, we introduce constraints that stem from the table geometry, i.e. avoiding hitting left, right, and robot's own band (the opponent's band is not reachable) and remaining on the table surface. The exact definition of all of these constraints is provided in Table 2, where $(x_{ee}, y_{ee}, z_{ee})$ is the end-effector position, $x_{ab}, y_{lb}, y_{rb}$ are the $x$ and $y$ coordinates of the robot's, left and right bands respectively, and $z_t$ is the $z$ coordinate of the table plane. To facilitate the exploration in the case of the PPO-Lag, TRPO-Lag and PCPO baselines we loosened the table height constraint to be a pair of inequality constraints that covers the range of $\pm 2$ cm around the original equality constraint. We also experimented with scaling up the entropy bonus to encourage exploration, however, it had no positive effect on the achieved performance.

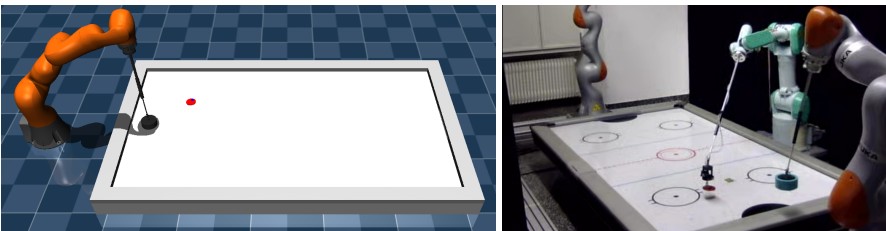

Figure 6: Visualization of the simulated and real-world Air Hockey Hitting task.

**Task definition.** In the air hockey setting the task definition vector $T$ is built using the initial observation of the environment. This vector contains the robot joint positions, joint velocities, puck position, puck orientation, and puck velocities. In this task, this task vector allows us to fully identify the desired trajectory, as the goal state is fixed, i.e., drive the puck into the goal area on the other side of the field.

### A.3 Air Hockey real robot deployment

As in the simulated setup, the real Air Hockey environment is composed of a Kuka Iiwa 14 robot arm (see Figure 6). The robot is equipped with an end effector composed of a metal rod, a gas spring, a passive universal joint, and a mallet. The mallet is composed of a movable attachment flange and the mallet itself. The flange is supported by a foam core, allowing for an additional compression of the end effector. This compression is particularly useful to avoid damage in case of small constraint violations and allows the robot to compress slightly the mallet on the table, reducing the probability of flipping. Furthermore, we have another Mitsubishi PA10 robot equipped with a suction cup, that is used to reset the puck position on the table.

The Kuka robot is controlled by an Active Rejection Disturbance Controller with a linear trajectory interpolator. The trajectory controller takes the desired position, velocity, and acceleration as input. The interpolator interpolates them linearly into a 1000 Hz command while the learned agent generates the command at 50 Hz. While this interpolation scheme does not produce realizable trajectories, the control frequency of 1000 Hz provides a smoothened command and avoids spikes in the interpolation, which are often problematic for high-order interpolation. A small safety layer is applied to the system to ensure that no wrong commands are applied to the system and that no command is skipped, causing a sudden stop of the commanded trajectory. This is to prevent damage to the robot, but cannot impose any of the safety constraints considered in this paper.

We use the Optitrack motion tracking system to track the puck at 120Hz. Different from the simulated environment, we block the goal areas to allow the PA10 arm to easily and automatically reset the puck after each hit attempt. We reset the puck in a predefined grid of positions. However, the airflow of the air hockey table causes the puck to drift randomly. This makes it impossible to evaluate exactly specific hitting positions with the automatic reset setup.

Table 2: Definition of the constraints in Air Hockey Hitting task.

| No. | Name | Definition |
|---|---|---|
| 1-7 | Joint positions | $|q| \leqslant [2.97, 2.09, 2.97, 2.09, 2.97, 2.09, 3.05]$ |
| 8-14 | Joint velocities | $|\dot{q}| \leqslant [1.48, 1.48, 1.75, 1.31, 2.27, 2.36, 2.36]$ |
| 15 | Robot's band collision | $x_{ee} > x_{ab} + r_m$ |
| 16 | Left band collision | $y_{ee} < y_{lb} - r_m$ |
| 17 | Right band collision | $y_{ee} > y_{rb} + r_m$ |
| 18 | Table height | $z_{ee} = z_t$ |

# B Bimanual manipulation task

In the main paper we presented an application of the proposed method in complex single-arm manipulation tasks. However, CNP3O is more general and may be successfully applied to wider ranges of tasks. To showcase this, we evaluated our solution in a different yet still complex task, i.e. bimanual manipulation.

## B.1 Task description

The Bimanual manipulation task is an adapted version of the task of moving an object handled by two UR5 manipulators and placing it in predefined position introduced in [61], where it was used as a benchmark for imitation learning. This task is implemented in simulation using MuJoCo and in Figure 7 we present its visualization. The main difficulties of this task is to move arms synchronously to do not drop or damage the object and place it very precisely such that pegs located in the desired position will fit the holes of the manipulated object. We make this task even more challenging by adding a preference to faster motions by setting the discount factor $\gamma = 0.997$.

A single episode is defined by the initial joint configuration of both arms and the orientation of the common base, which is computed using inverse kinematics based on the randomly drawn position of the manipulated object, such that both robots hold the object by handles. We fixed the set of possible initial positions to cover a relatively large space around the rack, i.e. $(x, y, z) \in [-0.5, 0.5] \times [0.2, 0.8] \times [0.2, 0.6]$, and we sample it uniformly. In turn, the desired pose of the object is fixed and is located 2.5cm centimeters above the stand, so that the pegs of the stand are in the holes of the object. The episode horizon is set to 400 3ms-long steps , which gives 1.2s for the entire episode.

In this task, the reward function is meant to encourage placing the handled object in the desired position, while avoiding dropping it. To achieve this we proposed the following reward function:

$$R = \frac{1}{100\frac{d}{d_{init}} + 1} - 0.01, \tag{4}$$

where $d$ is a weighted sum of the euclidean distance between the actual and desired position, and the angular rotation between the actual and desired orientation, with weights equal to 2 and 1 respectively. Similarly, $d_{init}$ represents the value of this distance at the beginning of the episode to normalize the rewards from different episodes. Moreover, to motivate reaching the goal while discouraging dropping the plate, which may easily happen especially when the holes do not fit the pegs on the rack correctly, we multiply the reward $R$ by 0 if the object is no longer handled by robots, and if object is successfully placed then we multiply it by 100. In both cases, we terminate the episode and to account for the remaining duration of the episode, we multiply the last reward by $\frac{1-\gamma^h}{1-\gamma}$, where $h$ is the remaining duration of the episode. We assume that the goal pose of the object is reached, when its weighted distance to the goal configuration is smaller than 0.015, which corresponds to the situation in which perfectly oriented object is already bedded in the pegs. In turn, dropping the object is defined as the gripper moving away from the gripping point by 1.5cm.

Moreover, also in this task, we identified safety and feasibility constraints, i.e. limited joint velocities and accelerations, as well as end-effector constraint, which ensures that the distance between the end-effectors is constant. This last constraint is needed to avoid squeezing and stretching of the object held. The exact definition of all of these constraints is provided in Table 3.

**Task definition.** In the bimanual setting the task definition vector $T$ is built using the initial state of the robots. As we always start from zero velocities, this vector consists of only initial robot joint configuration, which fully describes the given task, as the desired position of the manipualted object is constant.

## B.2 Experimental evaluation

In the considered task, we compared the performance of different motion primitives using CNP3O. We compared ProMP, ProDMP and two versions of the B-spline primitives. In all cases, we biased

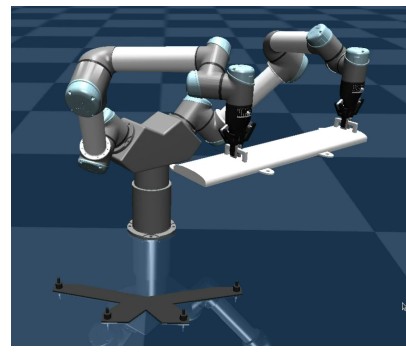

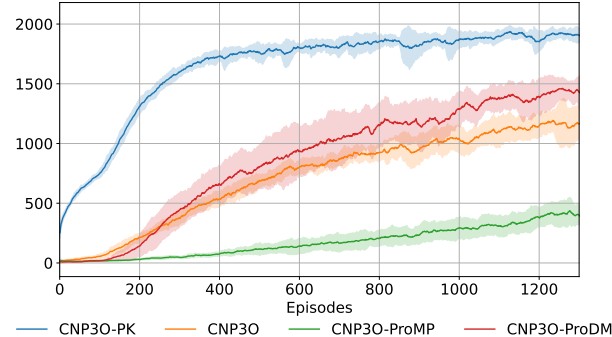

Figure 7: Visualization of the simulated bimanual manipulation task. The goal is to move white object to fit pegs of the black rack into its holes.

Figure 8: Learning curves (reward w.r.t. number of episodes) for the simulated bimanual manipulation task.

the end configuration of the arms to the solution found by inverse kinematics and end joint velocity to 0. However, this approach may be suboptimal, as at the end of the movement we want to achieve the velocity of the end-effectors and the object to move it down along the pegs. Therefore, in case of CNP3O-PK we enforced the end velocity to $0.2\,\mathrm{m/s}$ and end acceleration to $6\,\mathrm{m/s^2}$, such that we achieve relatively fast but also safe insertion (if acceleration mentioned above will be maintained, then the object stops after $33\,\mathrm{ms}$ covering a distance of $3.3\,\mathrm{mm}$ in this time).

In Figure 8 we present the learning curves for the considered motion primitives. One can see that by far the best results are achieved by the B-splines with prior knowledge about end velocity and acceleration, which showcase how important is the ability to impose boundary constraints in complex manipulation tasks. Among the remaining solutions, the best results are achieved by ProDMP. This confirms the usefulness of this motion primitive in the case of pick and place tasks. A similar but worse performance is achieved with B-splines, which offer greater flexibility along the whole trajectory hoever, due to this are not that effective in learning the 'go to' movements. In turn, the inability to enforce the end velocity of ProMP makes them struggle to learn efficient policy.

## C   Samples transformation function

To maintain the common scale of the standard deviations computed with $f_\theta^\sigma(T)$ and to put bounds on the sampled values, we introduced a transformation $\rho(\zeta)$ of samples $\zeta$.

Let's observe, that we may have several different ways to interpret the the MP weights. In the most common setting, they are just the weights of the configuration MPs. However, they may also be used to parameterize the time scaling factor $T_s$, control points of the time B-spline $r(s)$ or some boundary parameters, like desired position $\boldsymbol{q}_d$, velocity $\dot{\boldsymbol{q}}_d$ or acceleration $\ddot{\boldsymbol{q}}_d$. In this case, we may not want to have the same level of exploration noise for all of them. Moreover, we may impose some bounds on the predicted values, such that for example sampled velocity does not exceed the maximal one. To achieve this, we formulate the $\rho$ function for time-related samples by

$$\rho_t(\zeta) = \exp(a\zeta), \tag{5}$$

Table 3: Definition of the constraints in Bimanual manipulation task.

| No. | Name | Definition |
|---|---|---|
| 1-13 | Joint velocities | $|\dot{q}_i| \leqslant \pi$ |
| 14-26 | Joint accelerations | $|\ddot{q}_i| \leqslant 14$ |
| 27 | End effector distance | $\| \mathrm{EE}_{left} - \mathrm{EE}_{right} \| = 0.5$ |

where $a$ is a sample scaling factor. In the case of the weights related to configuration and its derivatives we transform the samples with

$$\rho_q(\zeta) = \Xi \tanh(a\zeta), \tag{6}$$

where $\Xi$ is the desired bound put on the sampled values.

In particular, in our experiments we have the following scaling factors: (i) for weights related to time $a_t = 1$, (ii) for configuration weights $a_q = 0.02$, (iii) for end configuration $a_{q_d} = 0.02$ in case of no prior knowledge, and $a_{q_d} = 0.007$ with prior knowledge, while $a_{\dot{q}_d} = 0.02$ and $a_{\ddot{q}_d} = 1$. Values of these scaling factors were chosen heuristically to achieve reasonable levels of initial exploration. In turn, the bounds $\Xi$ for the configuration-related weights are set to $\pi$ rad for the air hockey hitting task, and to $2\pi$ for the heavy object task. When weights are used to parameterize the desired velocity adjustment, then we use $2\dot{q}_{max}$ to allow for completely reversing the velocity bias, while in case of desired accelerations, we set them to $\ddot{q}_{max}$.

## D   Imposing prior knowledge on the trajectory representation

One of the big benefits of using MPs for the trajectory representation is the possibility to impose boundary constraints. In the most general formulation, we can impose the knowledge about the initial configuration on any type of MP. Let's note that if we have the trajectory defined by a MP, we can determine the first element of the weight vector $\boldsymbol{w}_1$, by solving a simple equation

$$\boldsymbol{w}_1 = \frac{\boldsymbol{\Phi}_{:,2:}(0)\boldsymbol{w}_{2:}}{\boldsymbol{\Phi}_1(0)}, \tag{7}$$

where index $2:$ means that we skip the first element. In the case of the basis functions with support equal to $[0;1]$, like ProMP this is theoretically all we can do. However, in practice, if the first basis function is very close to 0 for $s = 1$, then we can similarly compute the last weight, making only a very small error. To have this error equal to 0 one needs to have the support of at least one basis function to not contain 0, like in the ProDMP case. However, the most comfortable situation from the boundary conditions point of view is when the supports of the basis functions cover only some overlapping proper subsets of the domain, like for B-splines. In that case, we can fairly easily identify the subsequent boundary weights based on the boundary configurations and their derivatives. Then, if we can find the values of the next derivatives of the phase variable w.r.t. time, we can compute boundary configuration MP weights. Thus, we can impose the boundary conditions not only on the configuration but also on velocities, accelerations, and higher-order derivatives. For more details about computing them in the case of the B-spline trajectory representation, we refer the reader to [9].

In the next points we discuss what kind of prior knowledge about the task can be applied to different MPs, illustrating the usefulness of this feature and explaining in detail the structure of the models used for comparison in Figures 2, 3 and 10.

### D.1   Heavy Object Manipulation

Heavy object manipulation is an example of a task in which one of the main goals is to reach a certain pose of the manipulated object and maintain it till the end of the episode. In this type of task, typically many possible robot configurations satisfy the task objective. However, one may achieve significantly better results when biasing the end configuration with the one computed with inverse kinematics. We show this phenomenon in Figure 2 and 10, where all methods that utilize the prior knowledge outperformed their uninformed versions. In the considered task, the goal is not only to reach the target pose but also to stop the robot at that point. This kind of requirement cannot be directly imposed by both ProMP and ProDMP. However, the proposed B-spline MP allows one to set three last configuration MP weights in such a way that the last point of the trajectory reaches the goal with zero velocity and acceleration, which ensures smooth stopping.

### D.2 Air Hockey Hitting

In the air hockey hitting task, imposing the boundary conditions plays also a very important role. While in this case, it is not so obvious how to choose the desired final acceleration, we can provide a good initialization by setting a good hitting configuration and velocity. Using this technique, we can achieve significantly shorter training and obtain the highest rewards. In the experiments performed, we did this for the proposed CNP3O method, by biasing the end velocity and the hitting configuration with the values obtained with the optimization procedure proposed in [62]. Moreover, to compensate for the moving puck, we first computed the time B-spline to know the trajectory duration, and only then, using this duration for predicting the puck pose, we imposed the adjusted boundary conditions achieving decent hitting abilities from the very first episode. It is worth noting that incorporating the velocity level boundary conditions is not so straightforward in the case of ProMP and ProDMP. Therefore, the use of the B-spline-based MPs seems to be a better choice for tasks that require dynamic nonprehensile manipulation, like hitting the puck or tossing objects.

It is worth noting that the very important part of the air hockey hitting task, especially in terms of constraint satisfaction, is the stopping phase. After hitting the puck with a very high velocity robot needs to slow down its motion and at the same time still satisfy the table plane constraint, while potentially being in the state space areas of reduced manipulability. To handle this phenomenon, we leveraged the possibility of smoothly composing multiple trajectories. Instead of generating only the hitting motion we generate at the same time both hitting and stopping motion, which connects near to the hitting point with continuous accelerations.

### D.3 Choice of scaling factors

One of the key parameters of the black box policy optimization algorithm is the sample scaling factors, that take into account the desired level of the initial exploration.

While in general, this parameter can be set arbitrarily, in robotics settings these parameters are quite easy to tune. Indeed, the initial exploration in the task space should match the scale of the used robots and considered tasks. Therefore, given the initial values of the variance of the normal distribution generated by the neural network, we decided on the scaling factor for the MP weights. In the experiments for this paper, the desired end effector variance is in the order of decimeters.

However, some of the elements of the action space may have special meanings, like end configuration, velocity, or end acceleration. For these parameters, we may want to assign them different scaling factors keeping in mind that they represent different quantities, such as velocity and acceleration. These quantities may require different levels of exploration. For this reason, in our experiments, we decided to set a relatively high value for the acceleration— as we are very uncertain about its initial bias and the scale of the accelerations is much bigger than configurations —and a relatively low value for velocities— as we believe that the desired end velocity is very accurately given by the knowledge about the task. Similar reasoning was conducted also for the scaling factors associated with the time-scaling B-spline control points, but in this case, we focused on the desired level of exploration in terms of trajectory duration, which we wanted to be about 10% of the mean initial trajectory duration.

Thanks to the used parametrization, especially the B-spline-based one, the process of setting these parameters is quite intuitive due to the physical meaning of the considered quantities. Moreover, one can easily control the level of exploration by simulating a batch of random trajectories with the considered scaling factors and visually observing if the initial exploration matches the expectations.

## E Motion primitives flexibility

One of the goals of this paper was to bridge the gap between the SafeRL and MPs. Therefore, it is worth considering whether and to what extent we limited the generality of the classical CMDP framework by establishing a coupling with MPs.

We argue that the use of motion primitives does not introduce excessive limitations into the CMDP framework but rather limits the space of learnable policies and allows for an easy introduction of inductive biases. Nevertheless, there is an important difference w.r.t. classical CMDP methods, which is the trajectory-level (MPs requires black box learning) vs step-based exploration (classical CMDP). However, in general, there is no difference in terms of behavior we can achieve with black-box optimization (excluding the above-mentioned inductive biases) and classical CMDP formulation. The key difference is that step-based exploration exploits local information (possibly allowing faster learning) and black box exploration allows us to explore better, which is particularly useful in settings with sparse reward and complex constraints as the tasks presented in this paper, where classical CMDP methods are struggling. Finally, it is also possible to learn with MPs with classical CMDP methods, by adding Gaussian noise at the trajectory level, but this comes at the cost of the smoothness of the trajectory.

The main limitation that MPs, in general, introduces is the reduction in the space of learnable policies. Fortunately, this seems to not be very restrictive as in the literature we can see a broad range of applications in which MPs provided sufficient flexibility [3, 50, 63, 64, 65, 66, 67, 68]. In fact, the flexibility of every MP can be controlled by the number of basis functions used. However, the shape and distribution of the basis functions along the phase variable axis may affect the expressiveness of the particular MP. For example, ProDMP seems to be very effective in tasks where the final pose of the end-effector is particularly important, given the distribution of the basis functions and their unbalanced scale. Instead, the B-spline-based representation is characterized by flexibility along the entire trajectory and shines when we have complex constraints to impose due to the decoupling of the geometric path and the temporal path.

## F  Experimental details

While the general description of the performed experiments is included in the main text, we provide the details about them here to increase the reproducibility of our research.

First, as we mentioned in Section 2, CNP3O is meant to generate the trajectories, so it requires a controller to generate actions. In this paper, we used the proportional-derivative controller with feed-forward implemented in the Air Hockey Challenge repository [60] with default gains. The same controller is used by default by the ATACOM baseline. In turn, for the PPO-Lag, TRPO-Lag and PCPO we used the inverse dynamics algorithm from MuJoCo to transform the accelerations predicted by the policy into the torques.

The proposed method CNP3O utilize an episodic version of the PPO algorithm [53]. It is a direct analogue of the original algorithm, however, instead of adapting the policy based on the step rewards, it uses the rewards accumulated throughout the whole episode. Similarly, the value function estimate is computed only based on the task definition $T$. In practice, the computation of the task loss $L_{\text{task}}$ (see line 11 of Algorithm 1) for the given task $T$, sampled $\zeta$ and obtained discounted reward $J_{\text{task}}$ is defined by

$$L_{\text{task}} = -\min\left(\frac{f_{\theta_{new}}(\zeta|T)}{f_{\theta_{old}}(\zeta|T)}A_u, \text{clip}\left(\frac{f_{\theta_{new}}(\zeta|T)}{f_{\theta_{old}}(\zeta|T)}, 1 - \epsilon_{\text{PPO}}, 1 + \epsilon_{\text{PPO}}\right)A_u\right), \qquad (8)$$

where $\theta_{\text{new}}$ is the current value of $\theta$, while $\theta_{old}$ is the value of $\theta$ computed before current learning iteration, and $A_u = A - A_{\text{old}}$ is a standarized advantage, where $A = J_{\text{task}} - V_{\psi_{\text{new}}}(T)$ and $A_{\text{old}} = \frac{1}{N_{\text{episodes}}}\sum_{i=1}^{N_{\text{episodes}}}(J_{\text{task}_i} - V_{\psi_{\text{old}}}(T_i))$. Note, that $A_u$ is a constant through which we do not backpropagate the gradient.

Besides abovementioned differences we use the PPO implementation from MushroomRL library [69], however, instead of using the entropy bonus we utilized the entropy projection [70]. The entropy lower bound value was set to decrease linearly between the initial entropy lower bound $\nu_0$ and desired entropy lower bound $\nu_d$, which is meant to be achieved after $E_\nu$ epochs. After $E_\nu$-th epoch the entropy lower bound is to a constant value of $\nu_d$.

In Table 4, we present the architectures of all the neural networks used in the experiments, except the one used by CNP3O, as it is slightly more complex than a sequence of fully connected (FC) layers. In this case, the input is processed first by 3 FC layers, and then the resultant representation is used by two heads: (i) configuration head with 2 FC layers, and (ii) time head with a single layer. Each layer, except the output ones, consists of 256 neurons and a $\tanh$ activation function.

Table 4: Neural network architectures.

| Method | Hidden layers | Activation |
|---|---|---|
| Value network of CNP3O (all variants) | $4 \times 256$ | tanh |
| Policy network of CNP3O-ProMP/ProDMP | $4 \times 256$ | tanh |
| Actor network of ATACOM | $3 \times 128$ | SELU |
| Critic network of ATACOM | $3 \times 128$ | SELU |
| Actor network of PPO-Lag | $2 \times 256$ | tanh |
| Critic network of PPO-Lag | $2 \times 256$ | tanh |
| Actor network of TRPO-Lag | $2 \times 256$ | tanh |
| Critic network of TRPO-Lag | $2 \times 256$ | tanh |
| Actor network of PCPO | $2 \times 256$ | tanh |
| Critic network of PCPO | $2 \times 256$ | tanh |

Many of the chosen learning hyperparameters are common for both heavy object and air hockey hitting tasks, as well as for the considered methods. We list them in Table 5. However, some of the parameters are specific to the given algorithm, thus, we list them in Tables 6, 8, and 7.

Table 5: Common hyperparameters for all experiments

| Hyperparameter | Value |
|---|---|
| Number of episodes per epoch | 64 |
| Number of fits per epoch | 32 |
| Number of batches per fit | 1 |
| Batch size | 64 |
| Number of evaluation episodes | 25 |
| $\gamma$ | 0.99 |
| $\varepsilon_{\text{PPO}}$ | 0.05 |

Table 6: CNP3O hyperparameters

| Hyperparameter | Value |
|---|---|
| Number of configuration weights | 11 |
| Number of time weights | 10 |
| Initial standard deviation | 1 |
| Constraint learning rate $\alpha$ | 0.01 |
| Manifold metric decline bound $\beta$ | 0.1 |
| Policy learning rate (CNP3O all variants) | $5 \cdot 10^{-5}$ |
| Value function approximator learning rate | $5 \cdot 10^{-4}$ |
| Initial entropy lower bound $\nu_0$ | $n_b \cdot n_q$ |
| Desired entropy lower bound $\nu_d$ | $-n_b \cdot n_q$ |
| Entropy lower bound decline duration $E_\nu$ (air hockey) | 1000 |
| Entropy lower bound decline duration $E_\nu$ (heavy object) | 200 |
| Entropy lower bound decline duration $E_\nu$ (bimanual) | 500 |

## F.1 Learning duration

In the paper we presented the learning curves w.r.t. number of environment steps to visualize the sample efficiency of the compared approaches. However, it may also be worth to note how much time on average each of the method needed to complete an epoch of learning. We evaluated these times using a single core of Intel Core i5-12500H CPU and reported in Figure 9. The longest epoch duration is observed for ATACOM, as it performs policy updates much more often than the rest of the algorithms, the learning times of which are comparable.

Table 7: PPO-Lag, TRPO-Lag and PCPO hyperparameters

| Hyperparameter | Value |
|---|:---:|
| Actor learning rate | $5 \cdot 10^{-4}$ |
| Critic learning rate | $5 \cdot 10^{-4}$ |
| Lagrangian multiplier learning rate | 0.01 |
| Cost limit (heavy object) | 10 |
| Cost limit (air hockey) | 0.01 |
| $\lambda_{GAE}$ | 0.95 |
| Entropy bonus | 0 |

Table 8: ATACOM+SAC hyperparameters

| Hyperparameter | Value |
|---|:---:|
| Actor learning rate | $3 \cdot 10^{-4}$ |
| Critic learning rate | $3 \cdot 10^{-4}$ |
| Replay buffer size | $2 \cdot 10^{5}$ |
| Soft updates coefficient | 0.001 |
| Warmup transitions | $10^{4}$ |
| Learning rate of $\alpha_{SAC}$ | $5 \cdot 10^{-5}$ |
| Target entropy | -2 |
| Number of fits per step | 1 |

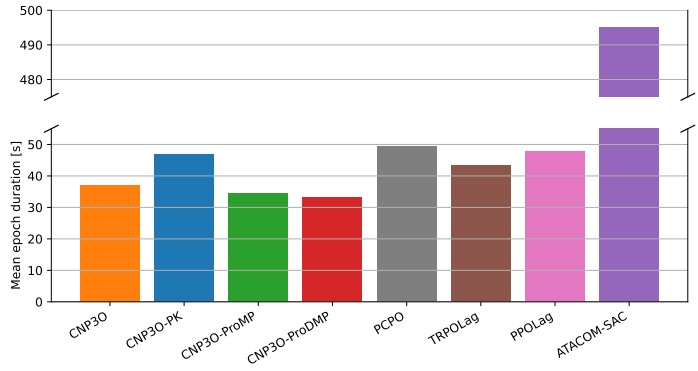

Figure 9: Mean learning epoch duration [s] in the Air Hockey task.

# G Additional results on the heavy object task

In this section, we present further analysis of the heavy object task, by looking at the metrics at the end of learning. We present in Figure 10 the boxplot showing the distribution of metrics of 100 episodes for each seed. Our results show that, in the setting with prior knowledge, the best parameterization for the motion primitives is the B-splines, achieving better performance and, in general, lower constraint violations than any other method. Adding prior knowledge is always an advantage for learning performance on all metrics, with the notable exception of B-Splines for collision penalty. However, it is worth noting that the collision penalty is extremely low compared to the other metrics, and most approaches can avoid collision robustly. Without prior knowledge, the B-spline parametrization obtains slightly worse constraint violations but also achieves slightly better performance. However, we remark that the B-splines can learn faster than the other motion primitives in this setting.

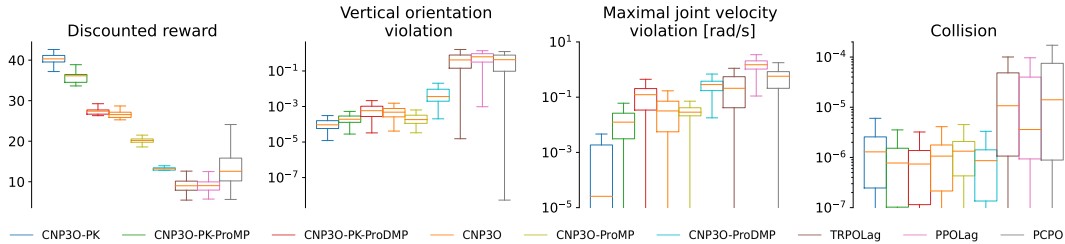

Figure 10: Statistical analysis of the considered approaches on the heavy object task

