# OpenReview forum: "Bridging the gap between Learning-to-plan, Motion Primitives and Safe Reinforcement Learning"
_robot-learning.org/CoRL/2024/Conference — CoRL 2024_

### Official Review · Reviewer_gN6n · 2024-07-20
**The paper tackles an important problem and has valuable theoretical insights but lacks a thorough and convincing experimental validation.**

**Originality:** 3
**Technical Quality:** 2
**Clarity Of Presentation:** 3
**Potential Impact:** 2
**Recommendation:** 2
**Confidence:** 3

**Review:**

The paper presents an approach to safe (constrained) reinforcement learning using robot control policies that are linear combinations of motion primitives from a given family. With the continuous advancements of learning-based methods and the importance of guaranteeing safety when deploying them in embodied systems, the tackled problem is indeed important and timely. The paper is sufficiently well-written even if some sections of the introductions and related work appear disjointed when discussing the (broad) space of safe robot control. It is a bit unclear what is the relation between CMDPs (as a formal framework) and motion primitives (as families of functions/trajectories): in the introduction, they seem to presented as competing approaches while in the methodology it appears that the latter is used as the family of functions where to look for a policy for the CMDP.
Although I am unsure about originality, I think some of the most interesting insights of the paper are in section 2.2, with regard to the choice of motion primitives. On the other hand, the experimental validation appears to be the weaker aspect of the paper: (i) the relatively narrow scope of the task being demonstrated does not allow to fully understand how general and useful the choice of the aforementioned motion primitives is; (ii) the very near perfect scoring ratios in simulation for both CNP3O-PK and the baseline ATACOM turning into 20% success ratio in real experiments (and in reverse place of the algorithms) makes it a bit dubious to draw definitive conclusions.

**Quality Of The Limitations Section:**

3

**Questions For Rebuttal:**

What, if any, are the relations between the choice of motion primitives, the type of task to be solved, the (unknown to the learning algorithm but not necessarily to the algorithm designed) robot dynamics, and the types of constraints?

How easily could this approach be ported to a robotic task other than single-arm manipulation? e.g. a collective task, or a mobile-robot task?

**Robotics Focus:**

2

**Summary Of Paper:**

The paper presents an approach to safe (constrained) reinforcement learning using robot control policies that are linear combinations of motion primitives from a given family.

**Summary Of Recommendation:**

I think that the paper could do a better job of explaining the connections between the chosen motion primitives, how much they limit (or not) the generality of the CMDP framework, and how specific to the tasks used in the experimental evaluation section.

---

### Official Review · Reviewer_j8eQ · 2024-07-21

**Originality:** 4
**Technical Quality:** 4
**Clarity Of Presentation:** 4
**Potential Impact:** 4
**Recommendation:** 3
**Confidence:** 3

**Review:**

Quality: Overall, the presentation of the paper is very clear and the paper is structured well. All the details of the experiments are provided and in detail analysis and comparison with other baselines is performed.

Originality: The paper proposes an algorithm for learning to generate trajectories in the motion primitive setting using known constraints and also compare their method to other MP based approaches. The paper also presents an approach for introducing flexibility in imposing boundary conditions using the time parametrization in the Motion primitive setup.

Significance:
The authors perform experiments on two challenging tasks and also show real world hardware experiments which demonstrates proof of concept of their method in real world. Their proposed method is faster and more stable during learning when compared to other motion primitive approaches.The author clearly addresses the limitation of the proposed framework and also lists ways to resolve some of the limitations.

**Quality Of The Limitations Section:**

3

**Questions For Rebuttal:**

No revisions needed.

**Robotics Focus:**

4

**Summary Of Paper:**

The paper presents an algorithm for learning to generate trajectories within the motion primitive framework using known constraints and compares their method to other MP-based approaches.

**Summary Of Recommendation:**

Please read the review above.

---

### Official Review · Reviewer_pgZh · 2024-07-23

**Originality:** 3
**Technical Quality:** 2
**Clarity Of Presentation:** 3
**Potential Impact:** 3
**Recommendation:** 3
**Confidence:** 3

**Review:**

Final Review:
I would like to thank the authors for addressing my concerns, and I appreciate the efforts put in during the rebuttal period. I couldn’t find the citation to PPO [1] in this paper, although it is a part of the algorithm.
I am inclined to change my recommendation to Weak Accept based on the additional details provided during the rebuttal period and because I find the contributions inspiring. However, I am still uncertain.

Minor remarks:

-Shouldn’t the 4th line of algorithm be \tau= \phi^T\rho(\zeta)

-Please provide implementation details of episodic PPO in the Appendix.

-The limitations regarding sample complexity of the algorithm are not entirely discussed in the paper. What is the training time of the algorithm compared to the baselines?

-Regarding “To facilitate the exploration in the case of the PPO-Lag, TRPO-Lag and PCPO baselines we loosened the table height constraint to be a pair of inequality constraints that covers the range of \pm 2 cm around the original equality constraint.” in line 572: Why wasn't the entropy bonus used, or if it was, why wasn't its coefficient increased instead of loosening the inequality constraints? It seems that loosening these constraints could reduce the performance of the baselines and potentially lead to constraint violations. However, I’m not entirely certain of the impact this would have, as increasing entropy bonus is not always effective. I suggest experimenting with different coefficients for entropy bonuses instead of loosening constraints if the paper is accepted.

[1]Schulman, J., Wolski, F., Dhariwal, P., Radford, A. and Klimov, O., 2017. Proximal policy optimization algorithms. arXiv preprint arXiv:1707.06347


--------------------------------
Strengths:
The paper presents a practical algorithm that integrates reinforcement learning (RL) with learning-to-plan techniques and MPs. Using MPs with constrained RL objective is novel.
It offers flexibility in imposing boundary conditions in Motion Primitives (MPs).
The robot air hockey experiment is interesting and demonstrates good zero-shot transfer.
It provides an analysis of different MP implementations for learning under constraints.

Weaknesses:

Although both PPO-Lag in [2] and the proposed CNP3O use tanh activation functions, the experiments for PPO-Lag were conducted using RELU activation. The rationale for selecting a different activation function should be discussed and empirically validated in the appendix.
Additionally, the paper lacks an explanation for not using Generalized Advantage Estimation (GAE), or if GAE was used, it is not mentioned.
The number of seeds used in the experiments is not specified.
Incorporating additional baselines [1] from other CMDP algorithms could enhance the reliability of the claims made in the paper.

[1]Gangapurwala, Siddhant, Alexander Mitchell, and Ioannis Havoutis. "Guided constrained policy optimization for dynamic quadrupedal robot locomotion." IEEE Robotics and Automation Letters 5.2 (2020): 3642-3649.

[2]Ray, Alex, Joshua Achiam, and Dario Amodei. "Benchmarking safe exploration in deep reinforcement learning." arXiv preprint arXiv:1910.01708 7.1 (2019):

[3]Schulman, John, et al. "High-dimensional continuous control using generalized advantage estimation." arXiv preprint arXiv:1506.02438 (2015).

[4]Andrychowicz, Marcin, et al. "What matters for on-policy deep actor-critic methods? a large-scale study." International conference on learning representations. 2021.

[5]Kicki, Piotr, et al. "Fast kinodynamic planning on the constraint manifold with deep neural networks." IEEE Transactions on Robotics (2023).

[7]Liu, Puze, et al. "Robot reinforcement learning on the constraint manifold." Conference on Robot Learning. PMLR, 2022.

**Quality Of The Limitations Section:**

3

**Questions For Rebuttal:**

Why are RELU activations used for PPO-Lag, given that both the original paper [2] and CNP3O use tanh?
How were the sample scaling factors determined in Appendix B?
Shouldn’t Algorithm 1 be wrapped with a for loop?
Shouldn’t trajectory tau be sampled from f_theta(T) in Figure 1 instead of tau = f_theta(T)?
Is GAE[3] used with CNP3O and PPO-Lag? If not, why?
How many seeds are used?
Could you provide more detail on the definition of the task with an example (input to the f_theta function) and the task distribution?

**Robotics Focus:**

4

**Summary Of Paper:**

This paper extends a learning-to-plan method, Constrained Neural motion Planning with B-splines (CNP-B)[5], to Reinforcement Learning(RL) for manipulation tasks with kinodynamic constraints. The proposed hybrid algorithm, Constrained Neural motion Planning with PPO (CNP3O), can generate motion primitives(MP)-based trajectories leveraging the known constraints without requiring full environmental knowledge. The method is evaluated in both real-world and simulation experiments, including a heavy object task and a robot air hockey task, showing promising results. The integration of learning-to-plan and reinforcement learning techniques demonstrates potential, but the experiments need further clarification and additional baselines.

**Summary Of Recommendation:**

The proposed method demonstrates potential, especially with its promising real-world implementation. However, there are a few areas that need further clarification to make a more solid recommendation. The baselines used in the paper could benefit from further elaboration and may not fully represent the current state of the art. More baselines [1] would help in better assessing the method's advantages. The motivation behind certain implementation details of the proposed approach and PPO-Lag seems missing. Specifying the number of seeds used in the experiments would also aid in evaluating the robustness and reliability of the results. Addressing these points during the rebuttal would strengthen the case for a positive recommendation. I am currently undecided and open to increasing my score.

---

### Author Rebuttal · Authors · 2024-08-08

Attached to this message, is the revised version of the paper. We highlighted in magenta the major changes in the paper. We thank the reviewers and the area chair for the insightful comments. We believe that the input of the reviewer has been very helpful in improving the quality of the paper.

---

### Decision · Program_Chairs · 2024-09-04

**Decision:**

Accept

**Comment:**

The paper proposes a practical and novel method that flexibly integrates RL with learning to plan techniques and motion primitives, demonstrates its zero-shot performance in an interesting air hockey experiment, and provides good analysis.

The reviewers have had important concerns about the performance of the method, the specificity of the task, and the used baselines, which may not reflect the state-of-the-art.

The authors provided detailed responses to the reviewers. Now the requested methodological and experimental details are included. More comparisons are made with SOTA. I recommend accepting the paper as a poster provided that the promised experiments are done by the authors.

The remarks of Reviewer pgZh should be incorporated.